# The Dynamics of Globally Unstable Air-Helium Jets and Its Impact on Jet Mixing Intensity

**Agnieszka Pawlowska** *[ID] **and Andrzej Boguslawski** [ID]

Department of Thermal Machinery, Faculty of Mechanical Engineering and Computer Science, Czestochowa University of Technology, 42-200 Czestochowa, Poland; abogus@imc.pcz.pl
* Correspondence: pawlowska@imc.pcz.pl

**Abstract:** The paper presents experimental investigations of the low-density air-helium jets. The paper is aimed at the analysis of the flow conditions promoting the local absolute instability leading to global flow oscillations. A number of the test cases are analysed with a wide range of the shear layer thickness showing conditions favorable for the global modes and also mixing intensity triggered by such a regime. It is shown that high mixing intensity is determined not only by the global regime but also by the vortex pairing process. The results are compared with a recently proposed universal scaling law for an onset into the global mode. The results turn out to be far from this scaling law and the reasons for such discrepancies are discussed. The measurements show also that if the shear layer at the nozzle exit is thin enough the global modes are suppressed. The mechanism leading to the global mode suppression under such conditions is carefully analysed.

**Keywords:** mixing; mixing length scales; turbulence

## 1. Introduction

A concept of absolute instability was first proposed by Landau [1] as a perturbation growing in time in contrast to the convective one in which the perturbation grows in space being swept away from its source by a convective stream. Such a phenomenon was first observed in plasma physics [2] and later it was shown theoretically by Huerre and Monkewitz [3] that it could be observed in simple shear flows. In the case of absolute instability perturbation growing in time travels downstream and upstream and contaminates the whole flow field. If a region of absolute instability is large enough it could lead to global self-sustained flow oscillations [4]. Monkweitz and Sohn [5] using spatio-temporal linear stability theory and Briggs-Bers criteria [2,6] showed that the absolute instability can be triggered in low-density axis-symmetric free jets provided that a ratio of the jet density to ambient fluid is lower than a critical value $S_{cr} \approx 0.7$. The main outcomes of the linear stability theory were confirmed experimentally in cardinal papers for the case with changing density by heated jet by Monkewitz et al. [7] and using helium-air mixture by Sreenivasan et al. [8] and Kyle and Sreenivasan [9], respectively. For both types of density changing significant oscillations were visible for the low-density cases. Using heated jet—studies reported by Monkewitz et al. [7], absolutely unstable modes were noticed for a density characterised by density ratio lower than the critical one equal to $S_{cr} = 0.65$. This type of oscillation was called Mode II. For the air-helium mixture, used by Kyle and Sreenivasan [9], the critical density of the oscillations appearance was lower than for heated jets and it was equal $S_{cr} = 0.61$ However, a characteristic feature of both cases was the presence of the axis-symmetric vortices and pairing of these structures. In general, the characteristic oscillations frequencies had a good agreement with the linear stability theory. There were also some differences in both cases. The presence of extra oscillations, called Mode I was characteristic for heated jets experiments exactly for density ratio $S < 0.69$. In turn, broadband oscillations were distinctive for air-helium jets provided

a very thin shear layer at the nozzle exit. The origin of the two described kinds of oscillations is still an open question. It is also puzzling that in another heated jet experiment, reported by Russ and Strykowski [10], even for as low density as equal $S = 0.5$, no global instability was detected. The significant mixing amelioration in the globally unstable low-density jet was detected in research upon concentration in self-excited helium jet reported by Richards et al. [11]. In turn, in momentum dominated helium jets experiment, performed by Yildirim and Agrawal [12], some buoyancy impact on oscillations frequencies was shown. As a complement of all prior experimental researches, a universal scaling law of the global mode frequency, based on the air-helium jet experiment, was formulated by Hallberg and Strykowski [13]. A universal law for the onset of global instability was formulated by Zhu et al. [14]. The global instability in helium jet using round and elliptical nozzles was studied experimentally by Tierney et al. [15].

Large Eddy Simulations (LES) and/or Direct Numerical Simulations (DNS) bring new insight into the understanding of low-density jets transition mechanisms. DNS predictions of low-density jets with a wide range of density ratio and shear layer thickness were performed by Lesshafft et al. [16] for the jet at $Re_D = 7500$. The global mode frequency predicted with DNS agreed well with the absolute frequency for the thickest shear layer at the global instability threshold density ratio. The frequency shift towards slightly higher values, of global mode relative to the absolute one, was observed for two groups of case types—for lower density ratio and for the thinner shear layer. However, the differences in the second group became more evident. It was also shown that vortex pairing phenomenon is characteristic only for a thin shear layer with ratio $D/\theta \geq 15$. LES predictions of the global mode in round low-density jet at $Re_D = 7000$ were presented by Foysi et al. [17] for the density ratio $S = 0.14$ and shear layer thickness $D/\theta = 27$. According to the authors, this was the first LES of a globally unstable round helium jet. They found excellent agreement with experimental data as far as oscillation frequency is concerned. They observed also the vortex pairing and side jets phenomena confirming the presence of global instability in the LES predictions. More recently, Boguslawski et al. [18] showed LES results of low-density jets analyzing an influence of the shear layer thickness and the velocity profile at the nozzle exit on the jet dynamics. By contrast to previous numerical results on low-density jets, they showed mean and fluctuating velocity in the near jet field. It was shown that intense velocity fluctuations of the order of 30% of the jet velocity are present for a low-density ratio $S < 0.3$. Global oscillations in the case of a higher density ratio, closer to the critical one, were equally high only if associated with the vortex pairing process observed for the sufficiently thin shear layer.

In most of the experimental studies of the low-density jets, the measurements were concentrated mainly on frequency characteristics of the globally unstable jet while experimental data concerning the velocity field and mixing intensity are still missing. The present paper is aimed at characteristics of both frequencies of the global oscillations and their influence on the jet mean and fluctuating velocity fields. The conditions of the experiment were also unique, for example, a very low level of disturbances at the nozzle exit, characterised by the turbulence intensity of the order of 0.05%. The experimental results presented were compared to two universal scaling laws: first one for the frequency the global modes, proposed by Hallberg and Strykowski [13], and the second for the onset into the global modes, proposed by Zhu et al. [14].

## 2. Experimental Set-Up and Measuring Apparatus

The vertical wind tunnel, shown in Figure 1, with the diameter of the nozzle exit $D = 15$ mm is considered to be an experimental stand. For needs of the experiment reported the rig characterised by a relatively high area contraction ratio of the nozzle was applied. The area contraction ratio was defined as a ratio of the inlet ($d$) and outlet ($D$) nozzle diameters in the following way $A = (d/D)^2$. The value of this parameter $A = 225$ was high enough to provide an extremely low level of turbulence intensity at the nozzle outlet about $Tu \approx 0.1\%$. The rig allowed the measurements at the range of Reynolds number from 5000 to 20,000 and the range of shear layer parameter $D/\theta = 24-162$ (where $\theta$—momentum thickness of the shear layer at the nozzle exit). The change in the layer thickness was

accomplished by applying cylindrical nozzle tips to the stand with lengths in the range $L/D = 1-25$, where $L$ was a tip length.

The air and the helium were supplied at the bottom of the set-up. Fans, filters and ducts were isolated from the measuring area and were located in the basement, not shown in the sketch. In the beginning, the medium passed through the pre-chamber with electric heaters (A) prepared for measurements in the heated air. Pre-chamber was filled with iron pellets (B) which increased the heat capacity and made the flow more uniform. Next, the medium entered the settling chamber (C) with seven wire gauzes (D) which also made the flow more uniform and reduced initial level of perturbation. In the station, there were two porous filters made of bronze, the first in the air supplying pipe (not shown here) and the second at the inlet to the settling chamber (E). Due to the very sensitive nature of the phenomenon investigated, with an extremely low initial turbulence level, the rig was placed on a thick plate mounted on vibro-insulators. The insulators reduced the impact of the external vibrations. The outlet of the nozzle was also surrounded by a wire gauze to prevent drafts and perturbations from natural convective motions in the laboratory. Helium was supplied perpendicularly to the direction of main air flow, directly after the first filter, which guaranteed a good mixing of helium with air and prevented backflow of helium. The helium stream was controlled using an air rotameter scaled in such a way that it was possible to read the helium flow rate.

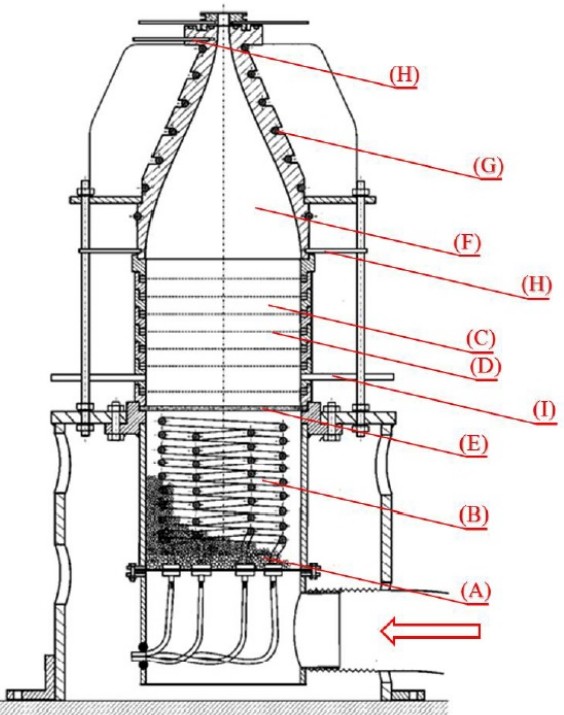

**Figure 1.** Sketch of the experimental rig.

A hot-wire method was used for measurements of the mean and fluctuating velocity fields in this variable-density flow. It was quite a controversial method due to the change in flow density along the jet, however, for each density the probe was calibrated in the flow with testing density. The measurement method can be considered appropriate because the phenomena occurring close to the nozzle outlet were still analysed in the region where the density was uniform. The same method was also used by Hallberg and Strykowski [13] for measurements of helium and nitrogen mixtures and by Zhu et al. [14] for helium-air jets. The measurements analysed in the paper were performed for two Reynolds numbers $Re = 5000$ and 10,000 with various lengths of nozzle tips resulting in a wide range of $D/\theta = 27.2 - 97.4$. The choice of the Reynolds number range was dictated by the ranges used in the cited experimental and numerical publications. Due to the very high aerodynamic drag of the rig,

related to the conditions ensuring an extremely low level of fluctuations at the outlet of the nozzle the tests carried out on the presented stand are limited to the Reynolds number Re = 20,000 for technical reasons. However, tests carried out on the stand showed that for such a high value of this parameter, the sought structures do not exist. Parameters of all the test cases analysed in the current experiment are gathered in Table 1 (where *H*—stands for the shape parameter of the shear layer at the nozzle exit).

**Table 1.** The shear layer parameters at the nozzle exit for all the test cases analysed.

| Test Case | L/D | Re | D/θ | H |
|---|---|---|---|---|
| $LD_1Re_5$ | 1 | 5000 | 70.6 | 2.596 |
| $LD_1Re_{10}$ | | 10,000 | 97.4 | 2.591 |
| $LD_7Re_5$ | 7 | 5000 | 41.8 | 2.544 |
| $LD_7Re_{10}$ | | 10,000 | 53 | 2.608 |
| $LD_{15}Re_5$ | 15 | 5000 | 30.2 | 2.502 |
| $LD_{15}Re_{10}$ | | 10,000 | 38.8 | 2.507 |
| $LD_{25}Re_5$ | 25 | 5000 | 27.2 | 2.588 |
| $LD_{25}Re_{10}$ | | 10,000 | 30.4 | 2.581 |

## 3. Linear Stability Theory Results

The experimental data were completed with sample spatio-temporal linear stability theory results, extracted from the work of Jendoubi and Strykowski [19] and Boguslawski et al. [18]. In both studies the base laminar flow velocity profile was approximated by the hyperbolic tangent function and the density distribution was derived from the Busemann-Crocco relation [20]. In the work of Jendoubi and Strykowski [19] the stability equations were solved by the shooting method while Boguslawski et al. [18] applied the spectral "tau" approach with the eigenfunctions approximated by the series of Chebyshev polynomials [21] to obtain complete maps of $\omega(k)$ ($\omega$-complex frequency, $k$-complex wavenumber). Then to find precisely the branch point $\omega^0$ for the absolutely unstable mode an iterative algorithm proposed by Monkewitz and Sohn [5] with the shooting method was applied.

## 4. Results

Figure 2 shows the mean and fluctuating velocity profiles along the jet axis for various density ratios and for Reynolds number $Re = 5000$ for the case with a relatively thin shear layer $D/\theta = 41.8$ (the test case $LD_7Re_5$). It can readily be seen that lowering the jet density caused a faster decay of the mean velocity associated with an increased level of fluctuations. However, from the results shown it is not possible to distinguish an onset to the global mode. It is known from the experimental studies [9] that in the case of global oscillations especially for low-density ratio $S < 0.3$ a peak of the fluctuations in the near jet field at the level of 30% of the jet velocity can be observed. For higher density ratio closer to the critical one, as pointed out by numerical studies [18], such a high level of fluctuations was always associated with the vortex pairing process. If the vortex pairing was not observed the level of the fluctuations was much lower despite that the jet was in the self-excited mode. Hence, a level of fluctuations cannot be considered as a single indication of the global mode. Global modes triggered by absolute instability are characterised by strong periodic vortices. Velocity spectra could be another tool to distinguish convective and global instability.

Figure 3 shows an evolution of the velocity spectra along the jet axis for the density ratio $S = 0.9$ and $Re = 5000$. For this density ratio, much higher than the critical one, the jet undergoes the classical convective Kelvin-Helmholtz transition. It can be seen a broadband peak emerging in the spectrum with the central non-dimensional frequency $St_D \approx 0.59$ at the distance $x/D = 3$ indicating the primary vortex development. Further downstream, at the distance $x/D = 5$, one can see much stronger oscillations with an order of magnitude higher amplitude. This is a distance, close to the end of the potential core, that is characterised by fully developed coherent vortices. At the distance of $x/D = 7$,

the amplitude is not higher than in the previous location but a subharmonic is emerging indicating the vortex pairing process leading further downstream to the vortex breakdown and fully developed turbulent flow. At the distance of the highest level of the fluctuating velocity $x/D = 9$ the flow is dominated by large scale coherent structures formed by the vortex pairing process characterised by the frequency $St_D = 0.59$.

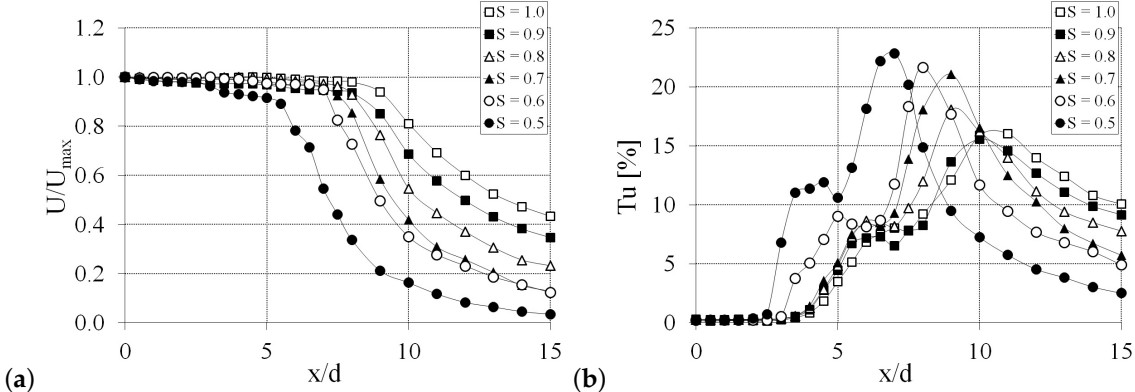

**Figure 2.** (**a**) Mean and (**b**) fluctuating velocity profiles along the jet axis, $L/D = 7$, $Re = 5000$.

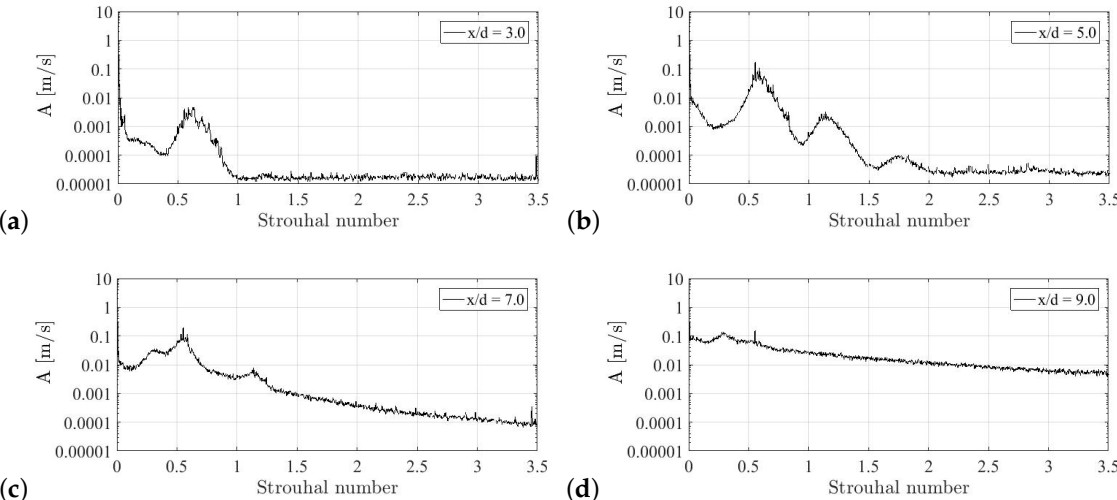

**Figure 3.** Evolution of spectra of axial velocity fluctuations, $S = 0.9$, $L/D = 7$, $Re = 5000$: (**a**) $x/D = 3$, (**b**) $x/D = 5$, (**c**) $x/D = 7$, (**d**) $x/D = 9$.

Figure 4 shows the mean and fluctuating velocity profiles for the test case $LD_7Re_{10}$. The flow dynamics in this case are different comparing to the previous case. The differences are caused by both the thinner shear layer and higher Reynolds number in this case. It can be seen that the velocity decays faster than in the case analysed previously and that the maximum of fluctuating velocity is closer to the nozzle exit. It is worth noting that for the smallest density ratio $S = 0.5$ a strong peak of the fluctuating velocity is observed at the distance $x/D \approx 4$. The level of oscillations exceeds 30% of the jet velocity as reported by Kyle and Sreenivasan [9] in helium jet. Spectra for this case and the density ratio $S = 0.9$, much higher than the critical one for an onset to global oscillations, are shown in Figure 5. As can be seen from Table 1 this case is characterised by the momentum thickness $D/\theta = 53$, slightly higher than for the lower Reynolds number. It is known from the linear stability theory [22] that a thinner shear layer leads to a higher frequency and growth rate of the Kelvin-Helmholtz mode. However, as can be seen in Figure 5 the situation, in this case, is slightly more complicated than for the lower Reynolds number and thicker shear layer. It should be noted that the shear layer thickness is now higher than the critical value $D/\theta = 50$ allowing triggering a self-sustained convective mode

reported by Boguslawski et al. [23–25] and Wawrzak et al. [26]. The self-sustained convective mode for $D/\theta = 53$ is characterised by the oscillations with the characteristic frequency $St_D = 0.75$ as shown by Wawrzak et al. [26]. It can be seen from Figure 5 that at the distance $x/D = 3$ two distinct peaks are seen in the velocity spectrum. The one with a higher frequency is characterised by $St_D = 0.7$, which means close to the frequency of the self-sustained convective mode, and the second one with a lower frequency $St_D \approx 0.6$ that can be interpreted as the Kelvin-Helmholtz mode. The existence of both peaks suggests that the Kelvin-Helmholtz mode and the synchronised self-sustained mode are present in the flow alternately. It is known from the numerical works [25,26] that the self-sustained convective mode is very sensitive to external disturbances and a hot-wire probe can easily suppress this instability regime. Indeed, further downstream at the distance $x/D = 4$ these two peaks merge into one broadband oscillation with growing a subharmonic indicating the vortex pairing process. Finally, at the distance $x/D = 5$ only the subharmonic is visible with the frequency of $St_D = 0.36$. Summing up the results obtained for the density ratio much higher than the critical one for global instability it is clear that under such conditions one deals with the classical convective Kelvin-Helmholtz instability. In the case of a lower Reynolds number and thicker shear layer one sees that the instability is characterised by a lower frequency and develops further from the nozzle exit than in the case of a thinner shear layer obtained for the Reynolds number $Re = 10,000$.

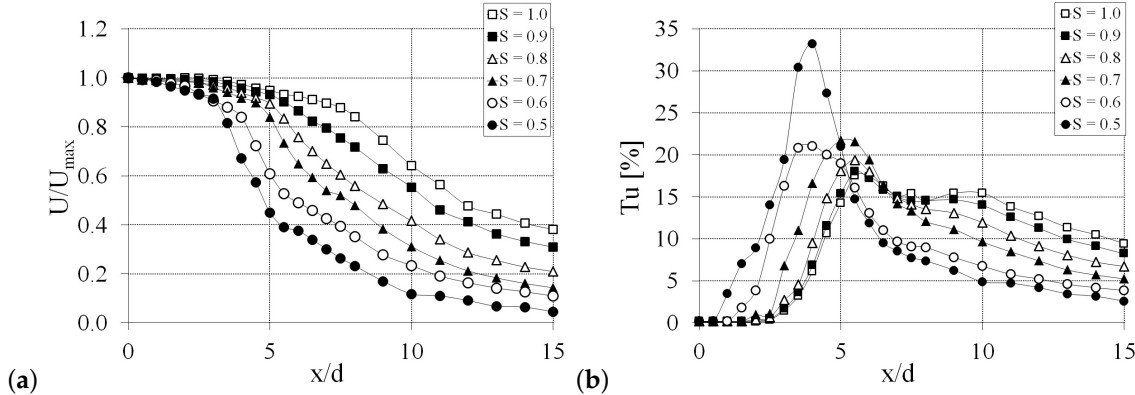

**Figure 4.** (**a**) Mean and (**b**) fluctuating velocity profiles along the jet axis, $L/D = 7$, $Re = 10,000$.

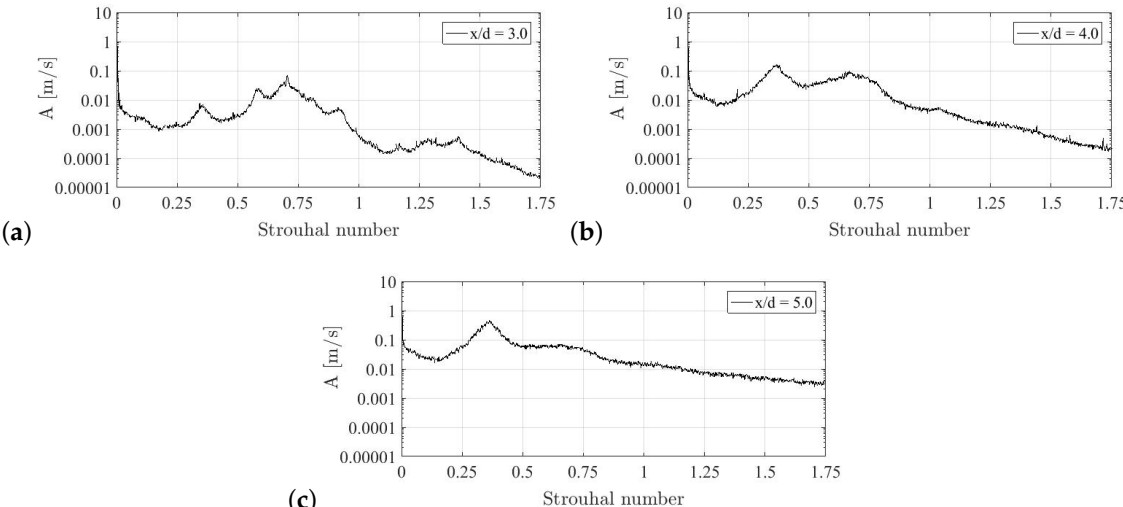

**Figure 5.** Evolution of spectra of axial velocity fluctuations, $S = 0.9$, $L/D = 7$, $Re = 10,000$: (**a**) $x/D = 3$, (**b**) $x/D = 4$, (**c**) $x/D = 5$.

Having known the flow dynamics with a density ratio above the critical value we will analyse the flow structure decreasing the density ratio to identify the onset into the global mode regime. Figure 6 shows boundaries between convective and absolute instability according to the linear stability theory compared with the onset scaling law proposed by Zhu et al. [14]. The solid line in Figure 6 corresponds to the results of the linear stability theory showed by Jendoubi and Strykowski [19]. They assumed the hyperbolic tangent velocity profile and the density distribution established with the Busemann-Crocco relation [20]. Boguslawski et al. [18] proposed a correction of this line assuming the Blasius velocity profile at the nozzle exit and a rectangular profile for the density (dashed line in Figure 6). Consequently, the density profile used in the linear stability calculations was much steeper than the velocity profile and shifted outward from the jet axis with respect to the velocity profile. The remaining lines correspond to the scaling law ([14]) for two Reynolds numbers used in the current experiment. Moreover, the points in Figure 6 show $D/\theta$ the parameters for all the test cases in the current experiment for the density ratio $S = 0.5$. It can be seen that according to the onset scaling law in the current experiment only three test cases correspond to the global instability conditions, namely the test cases $LD_1Re_5$ and $LD_1Re_{10}$ for the shortest extension tube and the thinnest shear layers for both Reynolds numbers and the test case $LD_7Re_{10}$. By contrast, using the linear stability theory with shifted density and velocity profile two additional test cases $LD_7Re_5$ and $LD_{15}Re_{10}$ are in the global mode region.

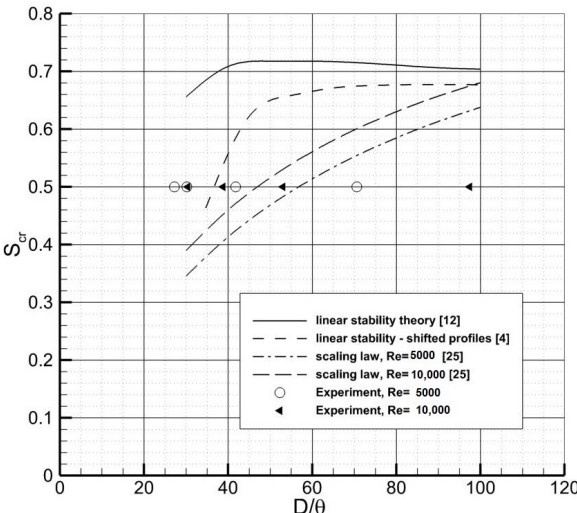

**Figure 6.** Convective/absolute/global boundary. Linear stability theory results of Jendoubi and Strykowski [19] and Boguslawski et al. [18] compared with a universal scaling law of onset into the global mode proposed by Zhu et al. [14].

Figure 7 shows mean and fluctuating velocity profiles for the density ratio $S = 0.5$ and $Re = 5000$ for all the extension tubes used in the current experiment. According to the onset scaling law only the test case $LD_1Re5$ is under the global mode conditions for $Re = 5000$. Looking at the fluctuating velocity profile one can observe that the profile for the thinnest shear layer is indeed very much different from the profiles for the cases $LD_7Re5$ and $LD_{15}Re5$ which could indicate a change of the transition regime. Moreover, looking more carefully at the fluctuating velocity profiles along the jet axis with a decreasing shear layer thickness it can readily be seen an exponential growth of the velocity perturbation in a range $x/D = 5 - 9$ characteristic for a development of the Kelvin-Helmholtz instability for the case $LD_{25}Re5$ that is not recorded in the cases with thinner shear layers. In these cases, the perturbations start growing closer to the nozzle exit for thinner shear layers but the growth rate is independent of shear layer thickness that could indicate global instability.

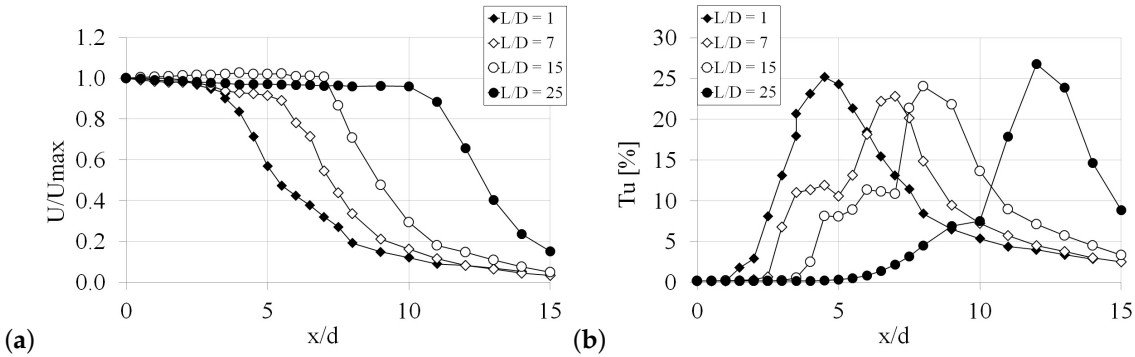

**Figure 7.** (**a**) Mean and (**b**) fluctuating velocity profiles along the jet axis, $S = 0.5$, $Re = 5000$.

A deeper insight into the flow dynamics requires analysis of the spectral content of the velocity oscillations. Figure 8 shows sample spectra for the flow with the density ratio $S = 0.5$ of the test case $LD_{25}Re_5$. Spectra are characterised with quite broadband oscillations similar to the spectra shown in Figure 3 for much higher density ratio indicating for the classical Kelvin-Helmholtz instability. However, in this case no traces of the vortex pairing process are observed due to a thick shear layer. A significant change of the spectra is observed for the test case $LD_{15}Re_5$ characterised by a thinner shear layer shown in Figure 9. In this case, one can see peaks with higher amplitudes within a narrow band of frequencies with many harmonics characteristic for the global instability. A similar spectra were obtained for the next test case $LD_7Re_5$ shown in Figure 10. In this case one can also see sharp peaks with many harmonics indicating the onset into the global modes. Significantly different spectra were observed in turn for the test case $LD_1Re_5$ shown in Figure 11. Further downstream from the nozzle at the distance $x/D > 3$ there is visible only one broadband oscillation without harmonics. It seems that an explanation of the mechanism of the transition in this case, characterised by quite a thin shear layer characterised by $D/\theta \approx 70$, can be found in the near jet field. Close to the nozzle exit at the distance $x/D = 2$, there are three peaks visible. The one with the highest frequency $St_D \approx 0.6$ is associated with its subharmonic. In between the primary peak and its subharmonic, there is an equally high peak characterised by the Strouhal number $St_D = 0.48$. It seems that in this region there are two different modes appearing alternately. The self-sustained convective mode $St_D \approx 0.6$ [23,25] for which very characteristic vortex pairing process close to the nozzle is manifested by the presence of subharmonic and the absolute mode characterised by $St_D \approx 0.48$. These two modes compete in the near jet field, however, further downstream large scale structures generated by the vortex pairing prevent absolute mode generation. It should be stressed that Kyle and Sreenivasan [9] observed also broadband flow oscillations in the helium jet characterised by a very thin shear layer. The results of the current experiment suggest that the origin of the broadband oscillations and suppression of the absolute mode could be the convective self-sustained mode which is present in jets with a thin shear layer. The rapid growth of the convective instability, in this case, can trigger a back-flow leading to the flow synchronisation [23] disturbing conditions for global mode.

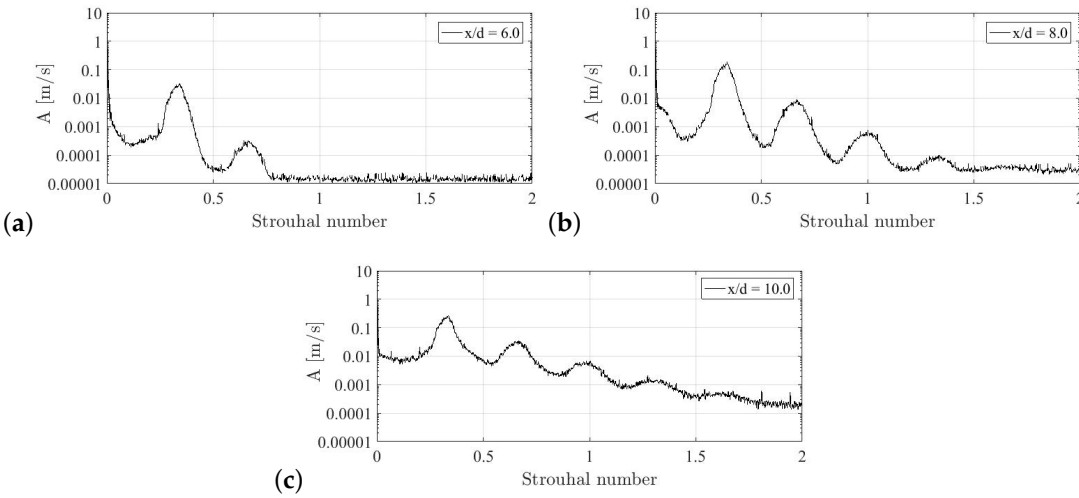

**Figure 8.** Evolution of spectra of axial velocity fluctuations, $S = 0.5$, $L/D = 25$, $Re = 5000$: (**a**) $x/D = 6$, (**b**) $x/D = 8$, (**c**) $x/D = 10$ .

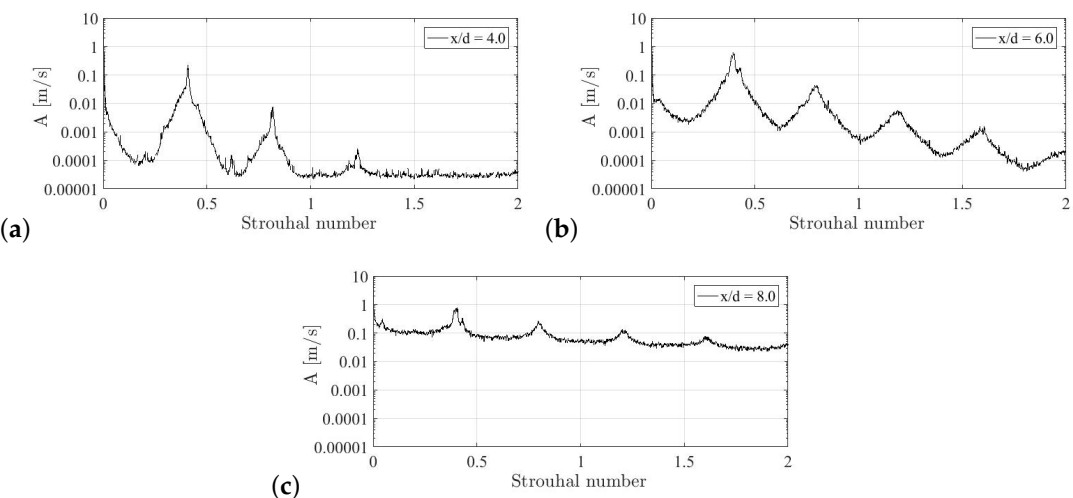

**Figure 9.** Evolution of spectra of axial velocity fluctuations, $S = 0.5$, $L/D = 15$, $Re = 5000$: (**a**) $x/D = 4$, (**b**) $x/D = 6$, (**c**) $x/D = 8$ .

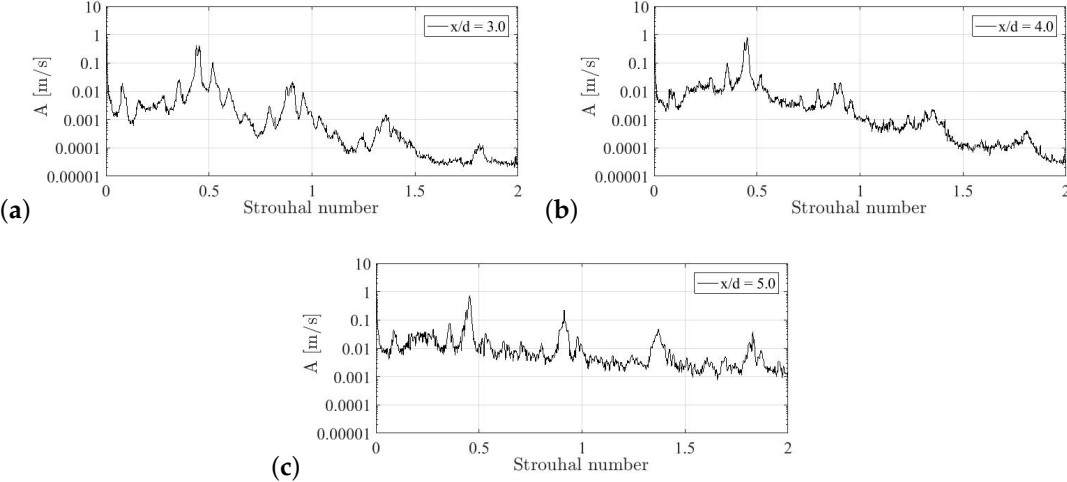

**Figure 10.** Evolution of spectra of axial velocity fluctuations, $S = 0.5$, $L/D = 7$, $Re = 5000$: (**a**) $x/D = 3$, (**b**) $x/D = 4$, (**c**) $x/D = 5$ .

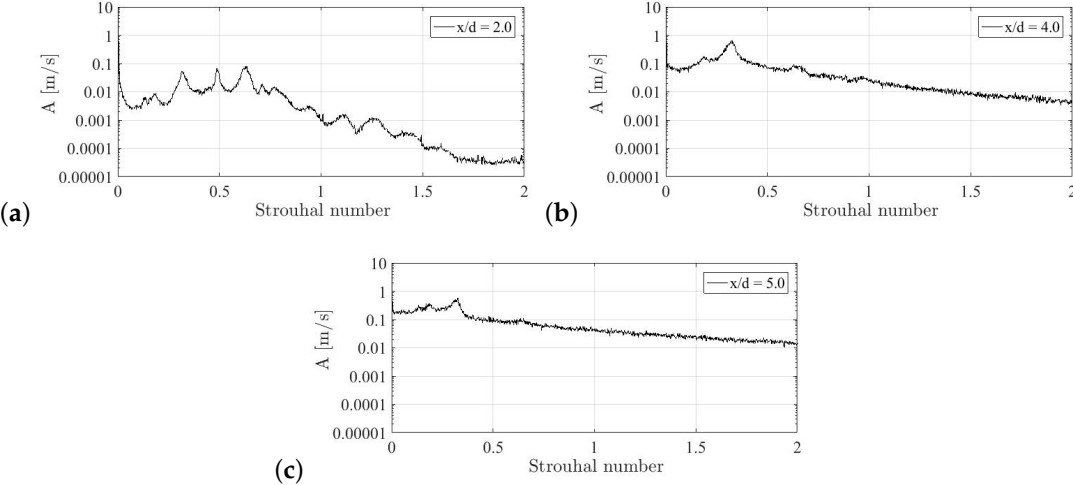

**Figure 11.** Evolution of spectra of axial velocity fluctuations, $S = 0.5$, $L/D = 1$, $Re = 5000$: (**a**) $x/D = 2$, (**b**) $x/D = 4$, (**c**) $x/D = 5$.

Similar measurements were performed for the same density ratio $S = 0.5$ but for a higher Reynolds number $Re = 10,000$. The mean and fluctuating velocities along the jet axis for these cases are presented in Figure 12. For the higher Reynolds number, an influence of the shear layer thickness is not so strong as in the previous case, however, some important differences can be observed. For the thickest shear layer in the test case $LD_{25}Re_{10}$ again an exponential growth of the fluctuating velocity starts at the distance $x/D \approx 2$ which is typical for Kelvin-Helmholtz instability. For the cases with thinner shear layers, the rapid growth of fluctuations starts more upstream and the growth rate is independent of the shear layer thickness what is typical for global modes. In the case $LD_7Re_{10}$ especially high oscillations are observed of the level of 35% of the jet velocity as observed for helium jets by Kyle and Sreenivasan [9]. In the case $LD_1Re_5$ a level of fluctuations is much lower. A confirmation of these observations based on the mean and fluctuating velocity fields can be found in spectra of the flow oscillations. Figure 13 shows spectra for the case $LD_{25}Re_{10}$. There are broadband oscillations around the $St_D \approx 0.4$ quite similar to the spectra shown previously for the test case $LD_{25}Re_{10}$. As in the previous test case we conclude that global oscillations were not detected. By contrast, in the two next test cases spectra of which are shown in Figures 14 and 15 sharp peaks undoubtedly indicate presence of the global mode. It is interesting to note that in the case $LD_7Re_{10}$ a strong subharmonic of the primary mode is visible already at the distance $x/D = 2$ manifesting the beginning of the vortex pairing process which further downstream leads to a very strong peak in spectrum at the distance $x/D = 4$ corresponding to the maximum of the fluctuating velocity. For the thinnest shear layer in the case $LD_1Re_{10}$ spectra of which are shown in Figure 16 it is evident again that the global oscillations are suppressed. Close to the nozzle exit at the distance $x/D = 1$ a peak with $St_D \approx 0.6$ is visible which could indicate the convective self-sustained mode which is associated with broadband oscillations around $St_D \approx 0.4$ probably resulting from an interaction of the subharmonic of the convective mode and the absolute mode. Further downstream only the broadband mode is present.

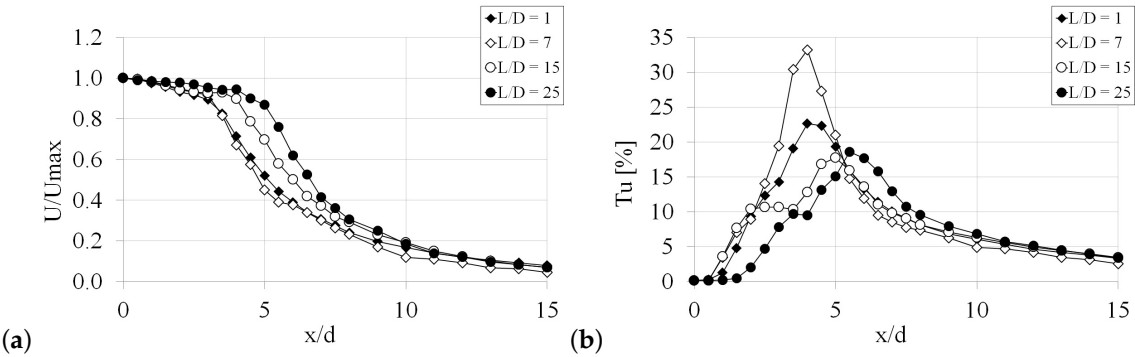

(a)  (b)

**Figure 12.** (**a**) Mean and (**b**) fluctuating velocity profiles along the jet axis, $S = 0.5$, $Re = 10,000$.

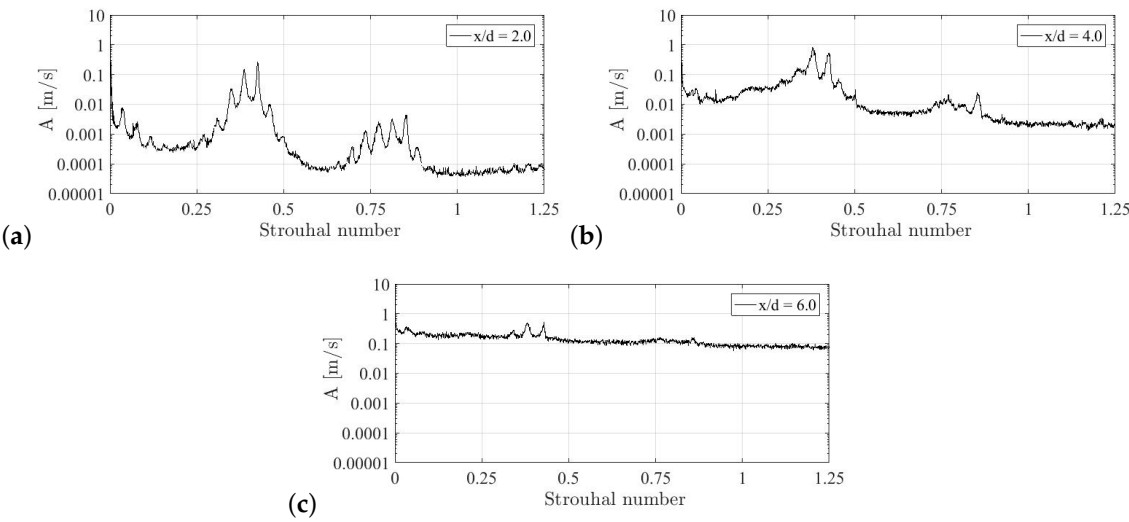

(a)  (b)

(c)

**Figure 13.** Evolution of spectra of axial velocity fluctuations, $S = 0.5$, $L/D = 25$, $Re = 10,000$: (**a**) $x/D = 2$, (**b**) $x/D = 4$, (**c**) $x/D = 6$.

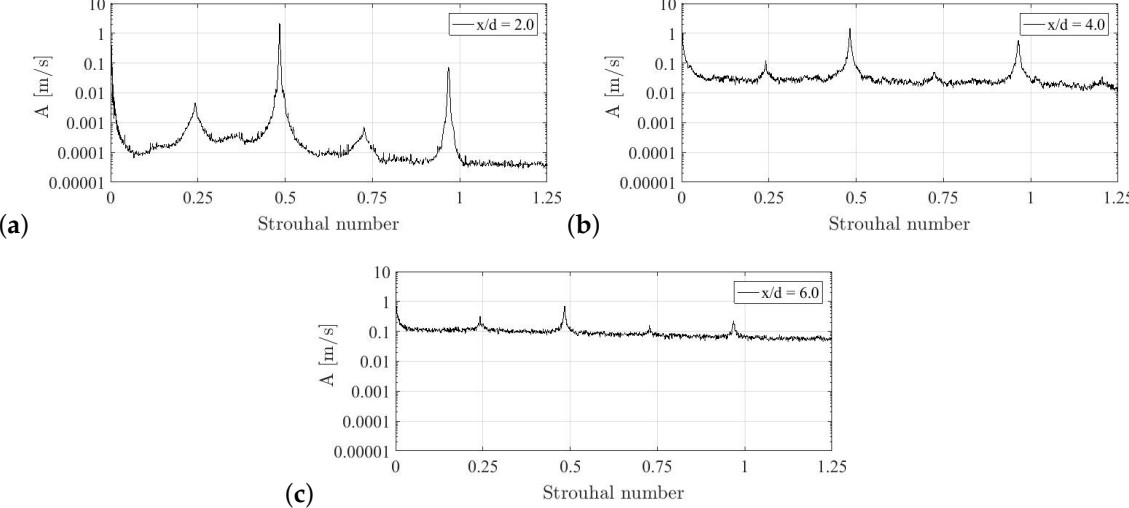

(a)  (b)

(c)

**Figure 14.** Evolution of spectra of axial velocity fluctuations, $S = 0.5$, $L/D = 15$, $Re = 10,000$: (**a**) $x/D = 2$, (**b**) $x/D = 4$, (**c**) $x/D = 6$.

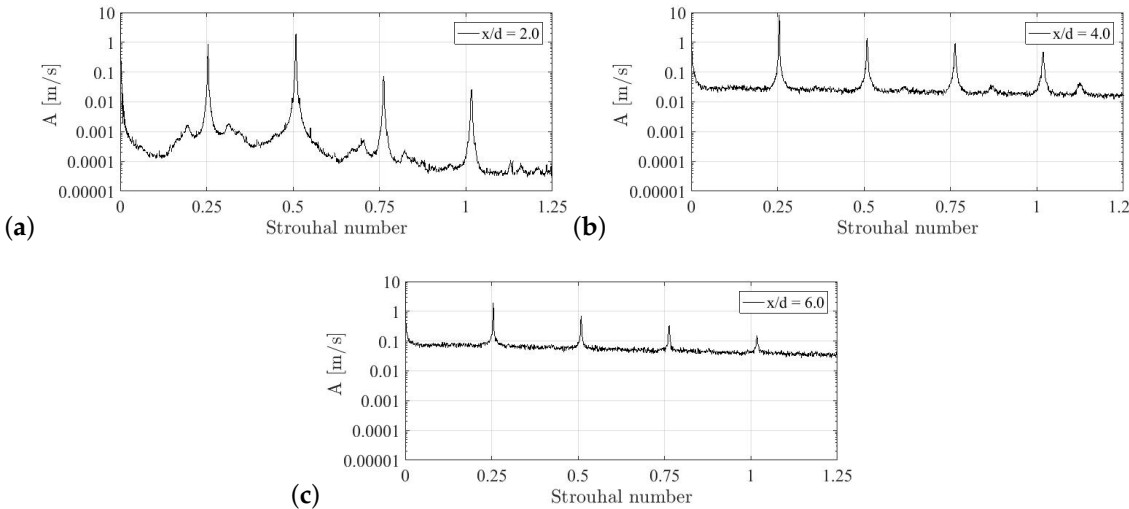

**Figure 15.** Evolution of spectra of axial velocity fluctuations, $S = 0.5$, $L/D = 7$, $Re = 10,000$: (a) $x/D = 2$, (b) $x/D = 4$, (c) $x/D = 6$.

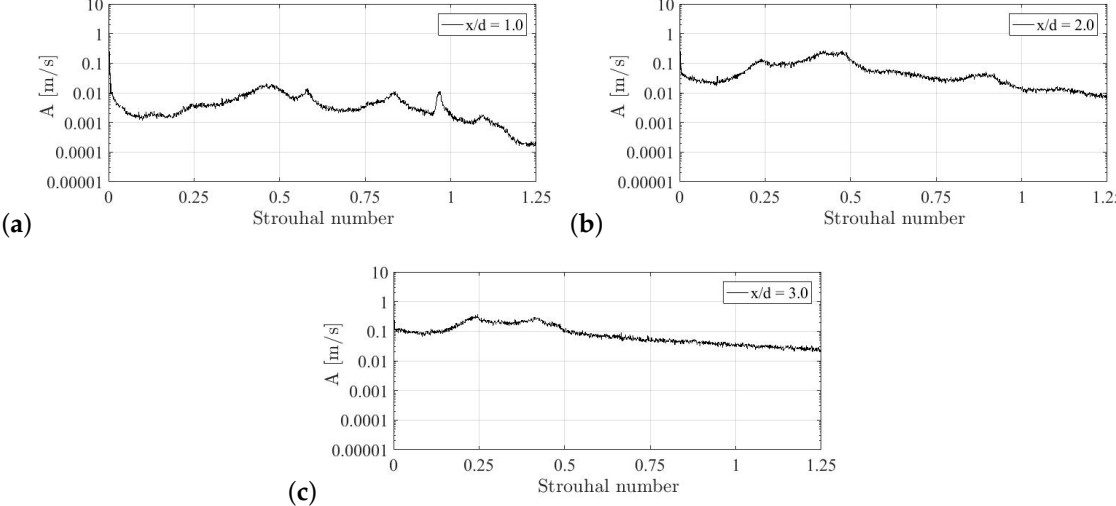

**Figure 16.** Evolution of spectra of axial velocity fluctuations, $S = 0.5$, $L/D = 1$, $Re = 10,000$: (a) $x/D = 1$, (b) $x/D = 2$, (c) $x/D = 3$.

Figure 17 shows non-dimensional frequencies of all the test cases analysed above indicating these ones that correspond to the global instability. It can be seen a monotonic growth of the frequency of the global mode as a function of the $D/\theta$ in qualitative agreement with the linear stability analysis and LES predictions shown by Boguslawski et al. [18]. The two cases with the thinnest shear layer namely $LD_1Re_5$ and $LD_1Re_{10}$ where broadband oscillations were detected are clearly of this line. Finally, all the test cases are confronted with the universal scaling law for the non-dimensional frequency of the global mode proposed by Hallberg and Strykowski [13]. Surprisingly, the results for all the test cases are located along the universal scaling law line (as shown in Figure 18). It shows that the detection of the global mode requires careful analysis of various flow characteristics and cannot be based on the comparison of the frequency with the universal scaling law.

Summing up the results discussed above for the Reynolds number $Re = 5000$ and the density ratio $S = 0.5$ the global instability was identified in the two test cases $LD_{15}Re_5$ and $LD_7Re_5$ that are characterised by $D/\theta = 30$ and $40$, respectively. As can be seen from Figure 6, according to the onset into the global mode scaling law [14], a required shear layer for $S = 0.5$ and $Re = 5000$ to release the global instability should be characterised by $D/\theta > 55$. For the Reynolds number

$Re = 10,000$ the universal scaling law indicates the critical shear layer thickness as $D/\theta = 45$ while in the current experiment the global modes were identified in the test cases $LD_{15}Re_{10}$ and $LD_7Re_{10}$ for which $D/\theta = 38.8$ and 53, respectively. According to the linear stability calculations (see Figure 6) for an inviscid flow with the density profile shifted outward with respect to the velocity one in the case of $S = 0.5$ the $D/\theta > 35$ leads to the absolutely unstable mode. A question arises why the current results are in contradiction with the onset law proposed by Zhu et al. [14]. The authors assumed that the universal scaling law for the onset into the global instability could be formulated by analogy to the scaling law for global mode frequency proposed by Hallberg and Strykowski [13] with the use of the density ratio, shear layer thickness and Reynolds number as the governing parameters. It seems that in the case of the onset into global instability one more parameter should be taken into account, which is unimportant as far as the global mode frequency is concerned, namely turbulence characteristics at the nozzle exit. One can easily imagine that for certain flow conditions the absolute instability could trigger the global modes if the flow at the nozzle exit is quiet enough while it is not possible at an increased level of inlet perturbation. In the current experiment due to a very high contraction in the nozzle and all the precautions applied in the settling chamber a turbulence level at the nozzle exit was very low, that could explain why the boundary between the convective and global instability was identified closer to the results of linear stability theory for an inviscid flow than to the scaling law proposed by [14] in which the level of perturbation at the nozzle exit was not defined.

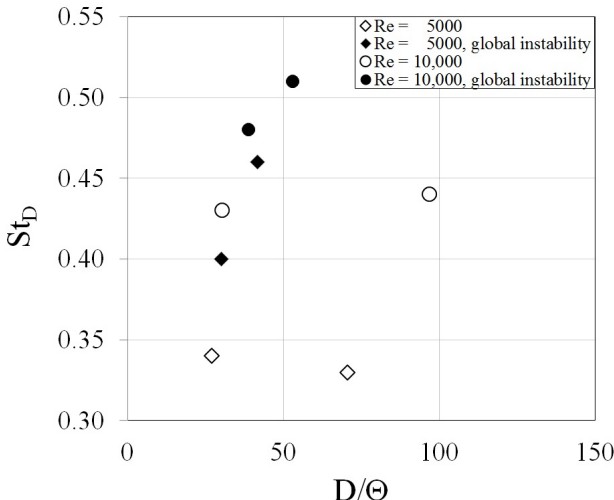

**Figure 17.** Non-dimensional frequency as a function of the $D/\theta$ parameter.

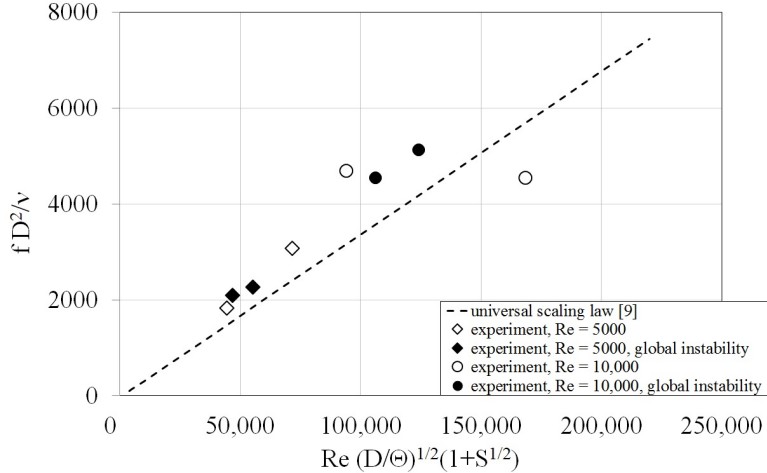

**Figure 18.** Experimental results vs. universal scaling law ([13]).

## 5. Concluding Remarks

The main goal of the research was to investigate the dynamics of the low-density air-helium jets aimed at an identification of the global flow oscillations stemming from the absolute instability. The global modes were identified looking at the fluctuations intensity, fluctuation growth rate and spectral distribution of the flow oscillations. It was shown that even that the jet is undoubtedly in the global instability regime the fluctuations level is not as high as for pure helium jets, however, if amplified by the vortex pairing phenomenon a very high mixing intensity can be observed even in the case of the density ratio close to the critical one. The results were compared with the universal scaling law for the onset into the global mode proposed by Zhu et al. [14]. It was shown that the onset into the global regime was possible for the shear layers much thicker than indicated by the scaling law. It seems that in a universal scaling law for the onset into the global mode the turbulence intensity at the nozzle exit should be added as an additional flow governing parameter. Finally, in the case of thin shear layer a suppression of the global modes was clearly shown, as reported by Kyle and Sreenivasan [9]. It seems that in the case of a sufficiently thin shear layer Kelvin-Helmholtz modes grow rapidly enough to create a back-flow that disturbs the flow at the nozzle exit distorting conditions for the absolute instability. The experimental studies performed shed new light on the dynamics of the low-density jets under global instability. It was clearly shown that mixing intensity controlled by the global modes depends significantly on the vortex pairing process leading to the fluctuations level higher than 30%. It was shown that for a sufficiently thin shear layer the global modes were suppressed. This if effect stemmed from the back-flow triggered by the Kelvin-Helmholtz instability that disturbed laminar flow at the nozzle exit. The results obtained coincide very well with the universal scaling low for the frequency of the global modes while some significant differences were observed with the scaling law for the onset into the global mode instability. It was concluded that universal scaling law for the onset into global instability has to take into account the turbulence level at the nozzle exit as an important governing parameter for low-density jet.

**Author Contributions:** Conceptualization, A.B.; methodology, A.B. and A.P.; software, A.P.; validation, A.P. and A.B.; formal analysis, A.B. and A.P.; investigation, A.P.; resources, A.P.; data curation, A.P.; writing—original draft preparation, A.B.; writing—review and editing, A.B.; visualization, A.P.; supervision, A.P.; project administration, A.P.; funding acquisition, A.P. The authors contributed equally to this work. All authors have read and agreed to the published version of the manuscript.

**Funding:** This research was funded by NATIONAL SCIENCE CENTRE, POLAND grant number 2016/23/N/ST8/03811—'Experimental verification of self-sustained oscillations in free round jet'.

**Conflicts of Interest:** The authors declare no conflict of interest.

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
