# Peer review of "The Dynamics of Globally Unstable Air-Helium Jets and Its Impact on Jet Mixing Intensity"

_processes, doi:10.3390/pr8121667_

Round 1

Reviewer 1 Report

Explain and justify the range of selected Re number
The explanation of linear stability calculations is too brief and needs to be further explained.
Justify and explain the use of the hyperbolic tangent function for the base laminar flow
Explain what made it necessary to use Chebyshev polynomials as opposed to other types of functions
Explain how did you solve the eigenfunction problem
Explain further details related to the branch point and its relation to absolutely unstable mode
Provide the linear equations for stability, and the form of the ansatz
Explain more details on Spatio-temporal linear theory
Explain in more details the large peak for S=0.7 in Fig 4

Reviewer 2 Report

The paper presents some interesting experimental investigations on a low-density air-helium jet. A good comparison with respect recently proposed universal scaling law for an onset into the global mode is reported into the manuscript. The work is quite interesting and probably deserves publication. The paper is moderating well written, anyway, some typos are present.

Introduction:
Literature survey looks well written with many references. This demonstrates that authors have important knowledge of the topic. Anyway, the last part of the introduction is not enough, is not clear what is the main novelty of the paper.

Experimental setup and linear stability calculation are well written.

The data analysis could be improved especially speaking about modes. A suggestion, probably for future works, is to use a time-frequency analysis.

Line 149, sentence: “It is known…..” authors should add a reference at the end
